# Multi-View Based Multi-Model Learning for MCI Diagnosis

**DOI:** 10.3390/brainsci10030181

**Published:** 2020-03-20

**Authors:** Ping Cao, Jie Gao, Zuping Zhang

**Affiliations:** School of Computer Science and Engineering, Central South University, Changsha 410083, China; ping.cao@csu.edu.cn (P.C.); gaojiexcq@csu.edu.cn (J.G.)

**Keywords:** alzheimer’s disease, magnetic resonance imaging, multi-view, CNN

## Abstract

Mild cognitive impairment (MCI) is the early stage of Alzheimer’s disease (AD). Automatic diagnosis of MCI by magnetic resonance imaging (MRI) images has been the focus of research in recent years. Furthermore, deep learning models based on 2D view and 3D view have been widely used in the diagnosis of MCI. The deep learning architecture can capture anatomical changes in the brain from MRI scans to extract the underlying features of brain disease. In this paper, we propose a multi-view based multi-model (MVMM) learning framework, which effectively combines the local information of 2D images with the global information of 3D images. First, we select some 2D slices from MRI images and extract the features representing 2D local information. Then, we combine them with the features representing 3D global information learned from 3D images to train the MVMM learning framework. We evaluate our model on the Alzheimer’s Disease Neuroimaging Initiative (ADNI) database. The experimental results show that our proposed model can effectively recognize MCI through MRI images (accuracy of 87.50% for MCI/HC and accuracy of 83.18% for MCI/AD).

## 1. Introduction

Alzheimer’s disease (AD), a neurodegenerative brain disease caused by multiple factors, is one of the most common chronic diseases in old age [1]. This disease usually causes progressive and disabling impairments of cognitive function, including memory, language, understanding and attention [2]. In 2015, it was estimated that about 47 million people worldwide had AD, and the number is expected to reach 141 million by 2050 [3]. At present, there is no practical method to cure AD [4], so early diagnosis of AD is needed to obtain treatment time. Mild Cognitive Impairment (MCI) is an intermediate state between normal aging and dementia [5], and one study showed that 32% of MCI converted to AD within five years [6]. Therefore, early diagnosis and intervention of Alzheimer’s disease is very important.

In the past few decades, neuroimaging has been widely used to study brain diseases [7,8,9]. Neuroimaging technology provides anatomical and functional images of the brain, such as Positron Emission Computed Tomography (PECT), Structural Magnetic Resonance Imaging (SMRI), Diffusion Magnetic Resonance Imaging (DMRI), Functional Magnetic Resonance Imaging (FMRI), Electroencephalogram (EEG), and Magnetoencephalography (MEG) [10,11,12]. Among them, SMRI is often used for the characterization and prediction of AD due to its relatively low cost and good imaging quality. Previous studies have shown that the volume and thickness of the brain are closely related to AD [13], the hippocampus region of AD patients is one third smaller than that of healthy subjects [14], and the medial temporal lobe region is the most effective region of the brain for identifying patients with MCI [15].

In recent years, machine learning and deep learning technologies have demonstrated revolutionary performance in many areas, such as action recognition, machine translation, image segmentation, and speech recognition [16,17,18,19]. Machine learning and deep learning have also achieved great success in the fields of medical image analysis and assisted diagnosis of brain diseases [20,21,22,23]. Unlike traditional methods based on manual feature extraction, deep learning can learn straightforward and low-level features from medical images, and construct complex high-level features in a hierarchical way [24].

The methods of diagnosing AD through Magnetic Resonance Imaging (MRI) images can be roughly divided into two categories, including 1) methods based on two-dimensional (2D) view, and 2) methods based on three-dimensional (3D) view. In most studies based on 2D view, 2D slices are selected from each subject, and these coronal, sagittal, or axial brain images are considered as a whole to classify. Bi et al. [25] manually selected brain images from three orthogonal panels of MRI data and performed unsupervised learning through PCA-Net. Neffati et al. [26] extracted 2D discrete wavelet transform texture features from coronal slices to classify AD. Jain et al. [27] selected the most informative set of 2D slices and used models trained on natural images to classify medical images through transfer learning. Mishra et al. [28] extracted features through complete local binary pattern from 2D slices in three directions instead of a single direction. These methods based on 2D view have all achieved excellent classification performance.

In addition, there are many methods based on 3D view that can also classify AD well [29,30,31,32]. Zhang et al. [33] discriminated between patients and healthy controls by using voxel-based morphometry (VBM) parameters from 3D images. Their results reported that the effect of classification using only 3D VBM parameters was better than the effect of classification using only 2D texture parameters. J. Liu et al. extracted 3D texture features through different divisions of the brain regions of interest (ROI) to construct multiple hierarchical networks [34,35]. Y. Wang et al. [36] diagnosed AD through combining morphometric measures of 3D images and connectome measures of 3D ROI. M. Liu et al. [37] used multivariate statistical tests to find discriminative 3D patch sets from the 3D rain images for the subsequent analysis. Basaia et al. [38] adopted a convolutional neural network to distinguish mild cognitive impairment who will convert to AD (c-MCI) and stable MCI (s-MCI) through 3D MRI images with an accuracy of 75%.

Considering the excellent performance of methods based on 2D view and 3D view, researchers in different fields have combined 2D and 3D views for study. Nanni et al. [39] combined texture features extracted from 2D slices with voxel features from 3D images and used multiple feature selection methods to improve the detection of early AD. In the field of action recognition and human pose estimation, Luvizon et al. [40] used a multi-task learning framework to combine 2D still images with 3D videos to learn features and achieved the latest results at that time. In the field of tumor segmentation, Mlynarski et al. [41] first divided 3D images into 2D images in three directions, and then fed them into the 2D model to generate 2D feature maps. Finally, they used 2D feature maps as the additional inputs of the 3D model to combine 2D and 3D information to achieve better segmentation results. The methods of combining 2D and 3D views in the different fields mentioned above have achieved excellent results. Therefore, we propose a multi-model learning framework based on multiple views to make full use of local and global information in this paper. We combine some 2D MRI images with the entire 3D MRI images, and adopt different deep learning models for different views to better distinguish AD and MCI.

The main contributions are listed as follows:(1)We employ entropy to select 2D slices and fuse them to learn 2D local features.(2)We propose a combination of 2D and 3D images of MRI to diagnose MCI, rather than a single view.(3)We propose a multi-model learning framework that uses different models to train data from different views.

## 2. Materials and Methods

We propose a new method of MCI diagnosis based on multi-view based multi-model (MVMM) framework. The MVMM framework mainly includes a 3D model for extracting global features and a 2D model for extracting local features. The 3D model we use is the Dilated Residual Network (DRN), which adds an Efficient Space Pyramid (ESP) module. The 2D model is the Dual Attention Inception Network (DAIN), which adds a dual attention mechanism to the Inception network. The flow of our MVMM framework is shown in Figure 1. Firstly, the gray matter (GM) images of subjects are divided into whole-brain gray matter images and some selected two-dimensional slices, and then they are respectively input into the corresponding different models. Finally, the local features and global features are concatenated together for integration training.

### 2.1. Data and Pre-Processing

#### 2.1.1. Data Acquisition

In this work, the MRI images we use are obtained from Alzheimer’s Disease Neuroimaging Initiative (ADNI) database (http://adni.loni.usc.edu/) [42]. ADNI started in 2004 under the leadership of Dr. Michael W. Weiner. ADNI is a longitudinal multicenter study with the primary goal of early detection of AD and the use of biomarkers to track disease progression. ADNI has already begun three phases, namely ADNI1 (2004-2009), ADNI2/GO (2010-2016) and ADNI3. At each phase of ADNI, new participants are recruited and agreed to complete various imaging acquisitions and clinical evaluations. Later phases include follow-up scans of some previously scanned subjects and scans of new subjects. In this paper, we select the SMRI data acquired at 1.5 Tesla [43]. The data we use are obtained from 649 subjects, which include 175 scans of AD, 214 scans of healthy controls (HC), and 260 scans of MCI. The demographic and clinical characteristics of all subjects are reported in Table 1.

#### 2.1.2. Data Pre-Processing

There is usually much noise in the raw data, so we need to preprocess the MRI data first. In this paper, we use the voxel-based morphological preprocessing method. Specifically, we use the CAT12 toolbox which is an extension to SPM12 [44] to provide computational anatomy. First, we register the MRI images to the standard space through DARTEL (Diffeomorphic Anatomical Registration Through Exponentiated Liealgebra) algorithm [45]. Second, we use the maximum a posteriori and partial volume estimation segmentation techniques [46] to segment the image into gray matter, white matter, and cerebrospinal fluid. Then, the Jacobian determinant is used to modulate the gray matter image nonlinearly. Finally, the gray matter image is spatially smoothed with the 8mm Gaussian smoothing kernel. The size of each gray matter image we get in the standard space is 121×145×121, then we use scikit-image package to resample it to a size of 96×96×96. It is noted that gray matter loss in the medial temporal lobe is characteristic of MCI [47], so we use gray matter images to analyze in this paper. The MRI images before and after preprocessing are shown in Figure 2.

### 2.2. DRN Model Based on a 3D View

For the whole brain three-dimensional view, we take the preprocessed whole gray matter image directly as input. In order to learn the 3D global information more comprehensively, we use 3D convolutional neural networks to perform global feature extraction related to AD. The 3D gray matter image contains the entire brain, which is very informative. Therefore, how to comprehensively learn useful features is a challenge. Convolutional neural networks have developed rapidly. Since the birth of AlexNet in 2012 [48], the depth of subsequent advanced convolutional neural network models has grown deeper. But as the depth increases, the problem of gradient disappearance during training becomes more serious. In order to avoid the problem of gradient disappearance caused by the network being too deep, the residual network (ResNet) [49] introduces an identity shortcut connection and skips one or more layers directly. Assuming that the layer l−1 is connected to the layer *l*, the output $x_{l}$ of the layer *l* is:(1)$$x_{l}=H_{l}\left(x_{l-1}\right)+x_{l-1}$$

H(·) represents a non-linear transformation function, including batch normalization (BN), rectified linear unit (ReLu), and convolution operation. ResNet reduces the difficulty of training deep networks by adding shortcut connections. For AD-related information, ResNet uses linear activation to obtain identity mapping. In contrast, ResNet uses non-linear activation for redundant information not related to AD. Since the non-linear activation is for redundant information, less useful information is lost. ResNet effectively solves the problem of network degradation caused by too deep depth, so that the model can learn more powerful advanced features.

In this paper, ResNet-18 is used as the basic model to train gray matter images from 3D view to obtain disease characteristics related to AD. However, the traditional ResNet model only uses 3×3×3 convolution kernels, and the receptive field of a single convolution kernel can only reach 27. In order to enable the convolution kernel to obtain a larger receptive field, and to allow the model to learn 3D features more comprehensively, we add the dilated convolution [50] on the basis of the ResNet network. Assuming that the size of the convolution kernel is *k* and the dilation rate is *r*, the receptive field (RF) of the dilated convolution is:(2)$$RF={\left[k+\left(k-1\right)\left(r-1\right)\right]}^{3},r>1$$

When r=1, it is ordinary convolution, and the receptive field is $k^{3}$. When r=2, for the same 3×3 convolution, the dilated convolution not only increases the receptive field (from 9 to 25), but also does not increase the parameters like ordinary convolution (the weight in the dilated part is 0). Although dilated convolutions increase the receptive field, the mesh effect is prone to occur if misused. Therefore, we use an efficient space pyramid (ESP) [51] module to avoid the mesh effect. Specifically, a 1×1×1 convolution is performed on the input to obtain a feature map of n×n×n×f, and then four parallel dilated convolutions are used. Finally, the hierarchical features are merged to obtain the same size feature map. The structure of the ESP module is shown in Figure 3.

In the figure above, the r1,r2,r3,r4 are different dilation rates. It can be seen that the residual operation is also used in the ESP module, and the information of different receptive fields is concatenated before the residual operation to ensure the output quality. We replace one layer in the original ResNet-18 network with an ESP module to learn the 3D global features of the brain image more effectively. The overall structure of the final three-dimensional DRN model is shown in Figure 1. The pre-processed whole-brain gray matter image contains much information. In order to effectively learn useful information, we choose 3D deep convolutional neural network for training. We use ResNet-18 with shortcut connections as our base model. At the same time, we also use dilated convolution to expand the receptive field in order to learn AD-related features more comprehensively.

### 2.3. DAIN Model Based on 2D View

For the 2D local view, we select some slices from the preprocessed gray matter image for training. We can select a large number of 2D slices from the 3D gray matter image. How to choose the best training data is very important for the success of the entire method. In this paper, we select 2D images based on image entropy [52] and extract the most informative slices to train the network. Generally, for a set of M gray values with probabilities $p_{1},p_{2},\ldots ,p_{M}$, the entropy can be calculated as follows:(3)$$H=-\sum_{i=1}^{M}p_{i}log_{p_{i}}$$

The higher the entropy, the more information the image contains. The subsequent problem is that MRI images generally contain much noise. Blindly selecting the image with a large amount of information may lead to the selection of some useless images with much noise. The images we study are preprocessed by CAT12. Compared with the original MRI image, the image we use is standardized and smoothed, and the skull of the image is removed. Therefore, we sort the slices in descending order of entropy and select the first 32 images for training to provide robustness according to previous research [53].

After obtaining the selected 2D slices from the 3D MR image, we use them as the inputs of 2D model to learn AD features. Assume that the image set of each subject is $x_{i}\left(x_{i}=\left\{x_{i1},x_{i2},\cdots ,x_{im}\right\}\right)$, the input of the two-dimensional model can be expressed as:(4)$$X=x_{1},x_{2},\cdots ,x_{n}$$

Thirty-two slices are selected from each subject through the axial brain image. In order to enhance the representation ability of the model, we use the Inception structure [54] to obtain information on different scales and fuse the features learned by convolution kernels of different sizes. In the 2D model, we pay more attention to the local information as the selected pictures are comparatively informative. How to find the local information that can distinguish MCI from other categories is the focus of our research. The attentional mechanism solves this problem by enabling the model to think globally and focus on more critical local information. We use a dual attention network (DAN) proposed by Fu et al. [55] that combines channel attention and spatial attention. The spatial attention module (PAM) in DAN first performs three 1×1 convolutions on the feature map *A* to obtain three feature maps *B*, *C*, and *D* of the same size (h×w×c). Moreover, these three feature maps are converted to the size of n×c
(n=h×w). Then the spatial attention map is calculated from *B* and *C*. We multiply the spatial attention map by *D*, and then multiply the result by the scale coefficient α (initialized to 0). Finally, the output of PAM is obtained by adding the original feature map *A*:(5)$$Attention_{pam}=\alpha {\left(softmax\left(C\cdot B^{T}\right)\right)}^{T}\cdot D+A$$

The channel attention module (CAM) in DAN first converts the feature map *A* into $A_{r}$, and multiplies $A_{r}$ and ${A_{r}}^{T}$ to obtain the channel attention map with the size of c×c. We multiply the channel attention map by $A_{r}$ and then multiply it by the scale coefficient β (initialized to 0). The result of the product is converted to the size of h×w×c. Finally, the output of CAM is obtained by adding to the original feature map *A*:(6)$$Attention_{cam}=\beta \left(A_{r}\cdot softmax\left({A_{r}}^{T}\cdot A_{r}\right)\right)+A$$

Finally, the results of the two attention modules are added together to form the output of DAN:(7)$$Attention_{out}=Attention_{pam}+Attention_{cam}$$

We combine DAN with Inception to form the final two-dimensional model. The structure of our proposed DAIN model is shown in Figure 1. The proposed DAIN model first obtains multi-scale AD information through the Inception structure. Then, the dual attention mechanism is used to obtain more significant local information. Finally, the important local information is combined with the fused multi-scale information to obtain the AD features that represent the 2D view.

### 2.4. Combination of 2D and 3D Views

Before constructing the multi-view model, we first integrate the output of the DAIN Model. Because in the previous DAIN model, we selected 32 slices for each subject and treated each slice as one subject. In this study, we use the above DAIN model as the pre-trained model of the final 2D model. In the final 2D model, the features extracted from each of the 32 images are concatenated together for classification. The final full connection is the MCI feature representing the local information extracted from the 2D slices.

After integrating the 2D features, we combine the features extracted from the 2D model with the features extracted from the 3D model. Specifically, 32 full connections of size 32 are obtained from every MRI image after the DAIN model. Then, through the final 2D model, the 32 full connections of the same subject are concatenated together to obtain the feature of size 32×32. After this layer of full connection, we add another full connection of size 32. That is, the information learned by the same person from the DAIN model is nonlinearly integrated, and then the integrated features are taken as the final two-dimensional AD features. The 2D model and the 3D model are trained separately. Finally, the full connections of the 2D model and the full connections of the 3D model are concatenated together for training. The method of combining 2D with 3D models is our proposed MVMM framework. Our MVMM framework can learn both global information of 3D images and local information of 2D slices.

## 3. Results

### 3.1. Experimental Setup

We implement the MVMM method through the Tensorflow and compute the model on the NVIDIA TITAN V GPU. The loss function we adopt is binary cross-entropy. We use He normal distribution to initialize the weights of the model. The learning rate of the 2D model is 0.001, while that of the 3D model and MVMM is 0.0001. The evaluation of the proposed method in this paper is based on the following two tasks:(1)T1: MCI/HC classification.(2)T2: MCI/AD classification.

To evaluate the classification performance, we adopt the 10-fold cross-validation strategy 10 times. We randomly select ten percent of the images from each class as the test set, and the remaining images are further randomly divided into 10 subsets for each category. In the process of cross-validation, each subset is taken as the validation set in turn, and the rest are used as the training set. This process is repeated ten times to get the final result. In this paper, the accuracy (ACC), sensitivity (SEN) and specificity (SPE) are used for evaluation. The three classification performance measures are calculated as follows:(8)$$ACC=\frac{TP+TN}{TP+TN+FP+FN}\times 100\%$$
(9)$$SEN=\frac{TP}{TP+FN}\times 100\%$$
(10)$$SPE=\frac{TN}{TN+FP}\times 100\%$$

Among them, TP represents true positive, FP represents false positive and FN represents false negative. In addition, we use the area under the receiver operating characteristic curve (AUC) to measure classification performance. The value with the best result for each measure is shown in bold in all tables of experiments.

### 3.2. Experimental Results

In this section, we present the respective results of the single-view and multi-view models, For the T1 task, it can be observed from Table 2 that the classification performance of MVMM model based on multiple views (87.50% for ACC) is better than the DRN model based on 3D view (81.67% for ACC) and the DAIN model based on 2D view (83.96% for ACC). For the T2 task, it can be observed from Table 3 that although the sensitivity of the MVMM is lower than that of the DAIN model, the ACC,SPE and AUC of the MVMM model (83.18%,70.56% and 0.8124) are higher than that of DRN model and DAIN model. Therefore, the performance of our MVMM model is the best. In addition, we perform Spearman’s Rank Order Correlation analysis to examine the relationship between the misclassification rate of MVMM model and the MMSE score of subject. As shown in Figure 4, the probability of MCI being misclassified as AD is negatively correlated with the MMSE score (*r* = −0.418, *p* = 0.033).

## 4. Discussion

In order to effectively verify the rationality of the model proposed in this article, we discuss it from the following six aspects: (1) selection of 3D models; (2) selection of 2D models; (3) selection of the number of 2D slices; (4) selection of combination methods; (5) performance on another dataset; (6) comparison with the whole-brain image without segmentation; (7) comparisons with related studies.

### 4.1. Selection of 3D Models

In this section, we compare the DRN 3D model used in this article with the VGGNet (VN) and DenseNet (DN) models. As shown in Table 4 and Table 5, VN and DN model perform poorly in both tasks (T1: 76.04% and 78.33% for ACC; T2: 75.91% and 76.59% for ACC). This is because the 3D MRI images contain a large amount of information, and direct use of the existing network structure cannot get a good effect. The basic model we selected is the ND-DRN model that does not use dilated convolution, which can reduce the redundancy of data by continuously adding shortcut connections in the learning process. Therefore, the ACC obtained by ND-DRN is better than the previous two models (T1: 79.38%; T2: 76.82%). Moreover, for more comprehensive learning, we add dilated convolution to expand the receptive field in the DRN model. Compared with the ND-DRN model, the final DRN model improves the accuracy rate by about 2% to 4%. The ACC,SPE and AUC of our DRN model in T1 and T2 tasks are the highest, while SEN is the second highest. Therefore, from a comprehensive perspective, our DRN model has the best classification performance.

### 4.2. Selection of 2D Models

As in Section 4.1, we compare the 2D model used in this article with the classic convolutional network models (AlexNet (AN) and MobileNet (MN)). It can be seen from Table 6 and Table 7 that the AUC of the AN and MN models is lower than the NA-DAIN model which does not use attention mechanism. Especially for the T2 task, the SPE of these two models are lower than 50%, which means that they cannot learn the deep local features of the 2D slices. The dual attention mechanism combines spatial and channel attention from a non-local perspective and makes full use of advanced brain features. In this way, more important information can be selected from the 2D data to diagnose MCI better. After adding the attention mechanism, the SPE of the DAIN model reaches 63.34%. At the same time, the ACC and AUC of the DAIN model are also the highest among these models (80.45% and 77.82%).

### 4.3. Selection of the Number of 2D Slices

In the two-dimensional model, we selecte 32 slices with the largest entropy from each gray matter image. In order to verify the effect of the number of slices on the model performance, we selecte 8, 16, 24, 32, 40, 48, and 56 slices from each gray matter image for comparison. It can be seen from Table 8 and Table 9 that the ACC of the model with eight slices is the worst (T1: 73.13%; T2: 70.68%), which means that too few 2D images cannot represent the brain information of the subject. When the number of slices gradually increases to 32, the ACC improves continuously (T1: from 73.13% to 83.96%; T2: from 70.68% to 80.45%). This indicates that the higher the number of 2D slices, the more disease-related information that can be learned. Nevertheless, it is worth noting that when the number of 2D slices exceeds 32, the ACC no longer improves. It can be inferred that the 32 slices are sufficient to represent the subject’s local brain information, and increasing the amount of information will lead to learning local information that is not related to the disease, resulting in a decrease in accuracy.

### 4.4. Selection of Combination Methods

After obtaining 2D local features and 3D global features, we combine DRN and DAIN for training. Table 10 and Table 11 demonstrate that the model has obtained good results, no matter if it is a combination of sum or concatenation. The concatenation fusion method we adopt combines 2D and 3D information from different spaces to achieve higher performance (accuracy of 87.50% for T1 and 83.18% for T2).

### 4.5. Performance on Another Dataset

In this section, we validate the generality of our proposed MVMM model on the OASIS (Open Access Series of Imaging Studies) dataset [56]. This dataset contains 3 or 4 individual MRI scans of 416 subjects aged 18 to 96. We use baseline MRI images of 198 subjects aged 60 or older, including 100 AD patients and 98 healthy controls. As shown in Table 12, we can also achieve good performance using the same preprocessing method and MVMM model on OASIS dataset. In other words, the MVMM model we proposed in this paper has potential to be used for wider research.

### 4.6. Comparison with the Whole-Brain Image without Segmentation

In this paper, the input of our model is the gray matter image after segmentation. In this section, we compare the classification performance using the whole-brain image (without segmentation) and using the gray matter image. It can be seen from Table 13 and Table 14 that all classification measures of the model using whole-brain images are lower than the model using gray matter images. This result is because whole-brain images contain more information than gray matter images, and we may need more data to learn the features associated with AD.

### 4.7. Comparisons with Related Studies

In this section, we perform experiments to compare the proposed method with other methods. The experimental results are reported in Table 15 and Table 16. Islam et al. [57] trained three 3D slices of each participant from the 2D view through ensemble learning. H. Wang et al. and Yuan et al. [58,59] studied the deep features of MRI from a 3D view through the novel convolutional neural network. Although they improved the MRI classification from 2D or 3D views, the verification results using the data in this paper are not as good as our proposed MVMM model. Our proposed MVMM model takes into account the extraction of 2D local information as well as 3D global information, which is more comprehensive than information extracted from a single view. Furthermore, we perform the Student’s t test on ACC of different methods to compare the performance. As shown in Table 15 and Table 16, our MVMM method is superior to other methods for all classification tasks at P<0.05.

## 5. Conclusions

In summary, we develop a multi-view based multi-model learning framework for the early diagnosis of Alzheimer’s disease. First of all, we comprehensively learn global information from the 3D view through residual networks and dilated convolutions. Then we perform the 2D view, which selects the most representative multiple slices through information entropy. Furthermore, we use the Inception network and dual attention mechanism to learn more crucial local information. Finally, we combine the models from 2D and 3D views to train for classification. In this paper, the deep features of MCI are studied both locally and comprehensively by MVMM learning framework. The experimental results show that the proposed method is effective and is expected to be used in the diagnosis of MCI. However, the training of different models in this paper is conducted in parallel, and the concatenation of features is performed at the end. In future research, more complicated combining methods can be considered to make full use of data for MCI diagnosis, such as combining texture features. 

## Figures and Tables

**Figure 1 brainsci-10-00181-f001:**
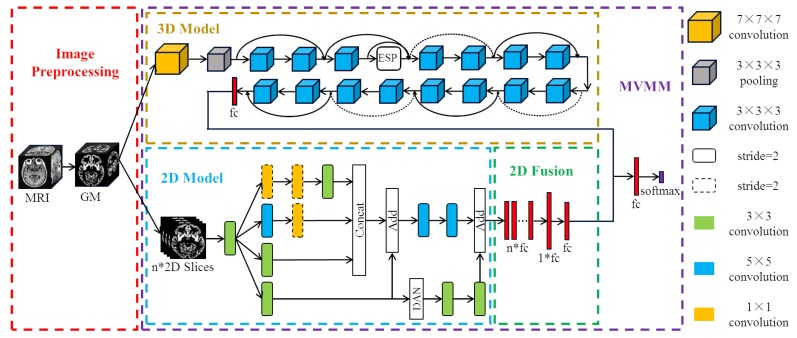
Multi-view based multi-model learning framework.

**Figure 2 brainsci-10-00181-f002:**
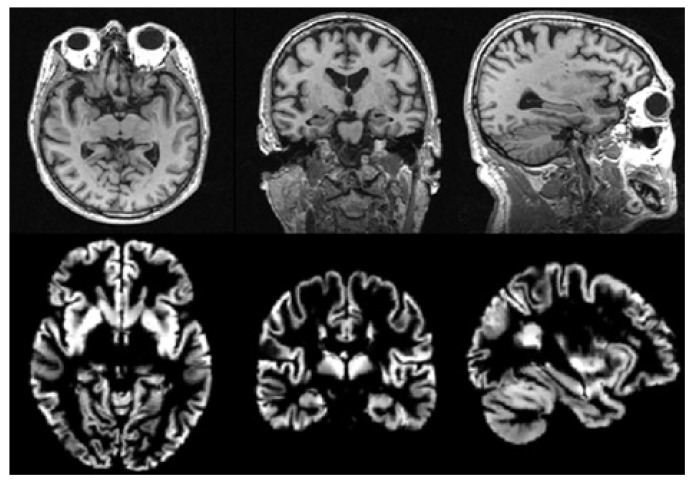
Images before and after preprocessing: The first row shows the original images, and the second row shows the final gray matter images.

**Figure 3 brainsci-10-00181-f003:**
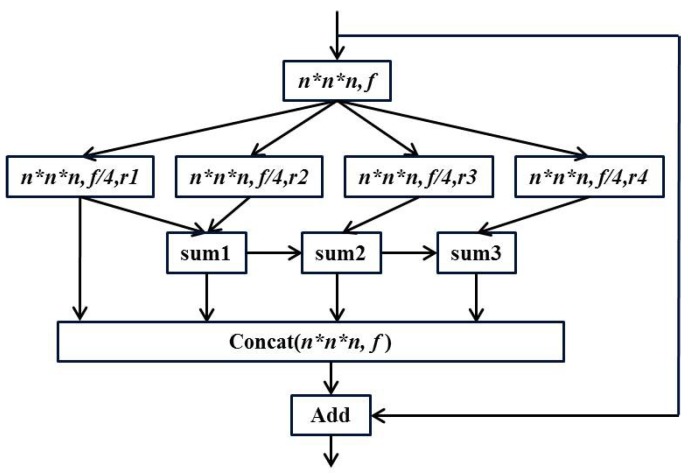
ESP module.

**Figure 4 brainsci-10-00181-f004:**
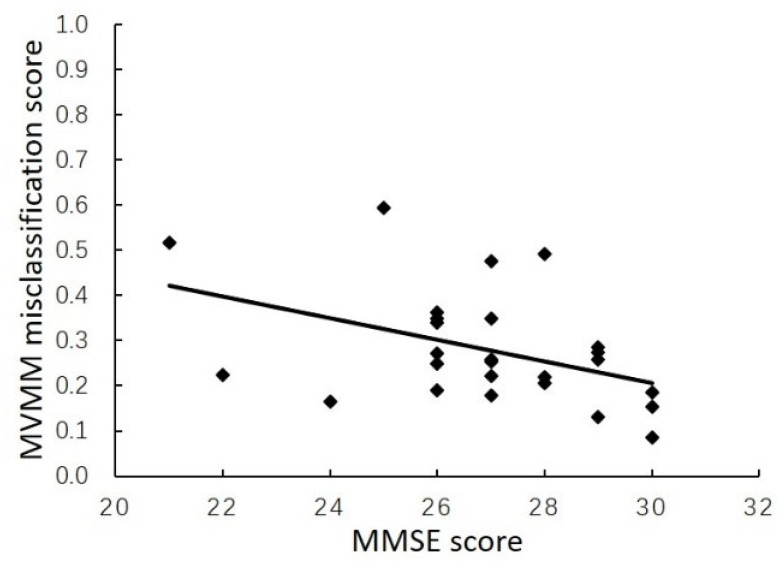
Relationship between the score of MCI being misclassified as AD and MMSE score.

**Table 1 brainsci-10-00181-t001:** Demographic information for 649 subjects.

Class	Subjects	Gender (M/F)	Age	MMSE
AD	175	93/82	75.62 ± 7.38	23.01 ± 2.67
HC	214	115/99	77.09 ± 5.21	29.16 ± 0.98
MCI	260	183/77	75.90 ± 7.37	26.14 ± 3.64

**Table 2 brainsci-10-00181-t002:** Classification performance of different views for T1 task.

Model	*ACC* (%)	*SEN* (%)	*SPE* (%)	*AUC*
DRN	81.67 ± 2.76	84.23 ± 3.53	78.64 ± 5.17	0.8143 ± 0.0311
DAIN	83.96 ± 2.32	87.31 ± 4.22	80.00 ± 3.63	0.8366 ± 0.0320
MVMM	**87.50 ± 2.08**	**89.62 ± 3.86**	**84.99 ± 3.39**	**0.8731 ± 0.0219**

**Table 3 brainsci-10-00181-t003:** Classification performance of different views for T2 task.

Model	*ACC* (%)	*SEN* (%)	*SPE* (%)	*AUC*
DRN	80.46 ± 3.08	89.62 ± 4.43	67.22 ± 6.23	0.7842 ± 0.0341
DAIN	80.45 ± 2.91	**92.31 ± 4.21**	63.34 ± 7.11	0.7782 ± 0.0326
MVMM	**83.18 ± 2.08**	91.92 ± 4.36	**70.56 ± 5.04**	**0.8124 ± 0.0248**

**Table 4 brainsci-10-00181-t004:** Classification performance of different 3D models for T1 task.

Model	*ACC* (%)	*SEN* (%)	*SPE* (%)	*AUC*
VN	76.04 ± 4.08	**84.62 ± 5.31**	65.90 ± 7.98	0.7526 ± 0.0454
DN	78.33 ± 3.75	79.23 ± 4.45	77.27 ± 5.75	0.7825 ± 0.0381
ND-DRN	79.38 ± 1.96	81.54 ± 4.69	76.82 ± 7.14	0.7918 ± 0.0270
DRN	**81.67 ± 2.76**	84.23 ± 3.53	**78.64 ± 5.17**	**0.8143 ± 0.0311**

**Table 5 brainsci-10-00181-t005:** Classification performance of different 3D models for T2 task.

Model	*ACC* (%)	*SEN* (%)	*SPE* (%)	*AUC*
VN	75.91 ± 3.08	86.54 ± 3.71	60.56 ± 5.67	0.7355 ± 0.0368
DN	76.59 ± 3.05	**91.92 ± 3.97**	54.44 ± 7.33	0.7318 ± 0.0423
ND-DRN	76.82 ± 2.98	89.23 ± 3.04	58.89 ± 5.96	0.7406 ± 0.0309
DRN	**80.46 ± 3.08**	89.62 ± 4.43	**67.22 ± 6.23**	**0.7842 ± 0.0341**

**Table 6 brainsci-10-00181-t006:** Classification performance of different 2D models for T1 task.

Model	*ACC* (%)	*SEN* (%)	*SPE* (%)	*AUC*
AN	74.37 ± 2.47	80.00 ± 6.84	67.73 ± 7.86	0.7388 ± 0.0243
MN	75.42 ± 4.54	77.69 ± 4.69	72.73 ± 7.34	0.7521 ± 0.0505
NA-DAIN	77.92 ± 3.39	80.00 ± 3.38	75.46 ± 4.09	0.7773 ± 0.0402
DAIN	**83.96 ± 2.32**	**87.31 ± 4.22**	**80.00 ± 3.63**	**0.8366 ± 0.0320**

**Table 7 brainsci-10-00181-t007:** Classification performance of different 2D models for T2 task.

Model	*ACC* (%)	*SEN* (%)	*SPE* (%)	*AUC*
AN	72.50 ± 2.95	90.38 ± 6.73	46.67 ± 9.15	0.6833 ± 0.0435
MN	69.99 ± 2.22	88.46 ± 5.95	43.33 ± 8.53	0.6593 ± 0.0262
NA-DAIN	75.46 ± 3.49	**96.92 ± 4.14**	44.44 ± 9.62	0.7068 ± 0.0422
DAIN	**80.45 ± 2.91**	92.31 ± 4.21	**63.34 ± 7.11**	**0.7782 ± 0.0326**

**Table 8 brainsci-10-00181-t008:** Classification performance of 2D models with different numbers of 2D slices for T1 task.

Number	*ACC* (%)	*SEN* (%)	*SPE* (%)	*AUC*
56	80.79 ± 5.56	86.16 ± 5.75	72.30 ± 3.72	0.7921 ± 0.0485
48	81.88 ± 2.29	84.62 ± 7.49	78.64 ± 8.83	0.8458 ± 0.0290
40	80.00 ± 4.28	83.85 ± 5.65	75.46 ± 5.20	0.7965 ± 0.0459
32	**83.96 ± 2.32**	**87.31 ± 4.22**	80.00 ± 3.63	**0.8366 ± 0.0320**
24	79.17 ± 5.51	77.69 ± 7.25	**80.91 ± 7.49**	0.7930 ± 0.0562
16	75.83 ± 3.97	80.77 ± 8.60	70.00 ± 6.18	0.7539 ± 0.0365
8	73.13 ± 4.41	78.08 ± 7.70	67.27 ± 8.57	0.7267 ± 0.0444

**Table 9 brainsci-10-00181-t009:** Classification performance of 2D models with different numbers of 2D slices for T2 task.

Number	*ACC* (%)	*SEN* (%)	*SPE* (%)	*AUC*
56	73.64 ± 3.08	91.54 ± 6.61	47.78 ± 6.18	0.6966 ± 0.0281
48	74.32 ± 2.69	91.15 ± 5.96	50.00 ± 8.95	0.7058 ± 0.0312
40	75.78 ± 4.72	92.16 ± 4.50	52.00 ± 9.67	0.7208 ± 0.0525
32	**80.45 ± 2.91**	**92.31 ± 4.21**	**63.34 ± 7.11**	**0.7782 ± 0.0326**
24	74.09 ± 3.96	80.38 ± 5.66	52.78 ± 8.32	0.7083 ± 0.0460
16	72.27 ± 3.34	85.39 ± 5.09	53.33 ± 9.02	0.6936 ± 0.0373
8	70.68 ± 3.43	86.15 ± 5.57	48.33 ± 5.58	0.6724 ± 0.0440

**Table 10 brainsci-10-00181-t010:** Classification performance of different combination methods for T1 task.

Model	*ACC* (%)	*SEN* (%)	*SPE* (%)	*AUC*
SUM	86.46 ± 3.12	85.77 ± 4.71	**87.27 ± 3.96**	0.8652 ± 0.0299
CONCAT	**87.50 ± 2.08**	**89.62 ± 3.86**	84.99 ± 3.39	**0.8731 ± 0.0219**

**Table 11 brainsci-10-00181-t011:** Classification performance of different combination methods for T2 task.

Model	*ACC* (%)	*SEN* (%)	*SPE* (%)	*AUC*
SUM	82.05 ± 2.95	**93.08 ± 5.10**	66.11 ± 7.03	0.7959 ± 0.0333
CONCAT	**83.18 ± 2.08**	91.92 ± 4.36	**70.56 ± 5.04**	**0.8124 ± 0.0248**

**Table 12 brainsci-10-00181-t012:** Classification performance of MVMM on OASIS dataset.

Dataset	*ACC* (%)	*SEN* (%)	*SPE* (%)	*AUC*
OASIS	85.65 ± 2.67	85.90 ± 3.83	85.00 ± 3.70	0.8540 ± 0.0291

**Table 13 brainsci-10-00181-t013:** Classification performance of models with different inputs for T1 task.

Input	*ACC* (%)	*SEN* (%)	*SPE* (%)	*AUC*
whole-brain	83.33 ± 2.46	84.23 ± 5.55	82.27 ± 4.57	0.8325 ± 0.0252
GM	**87.50 ± 2.08**	**89.62 ± 3.86**	**84.99 ± 3.39**	**0.8731 ± 0.0219**

**Table 14 brainsci-10-00181-t014:** Classification performance of models with different inputs for T2 task.

Input	*ACC* (%)	*SEN* (%)	*SPE* (%)	*AUC*
whole-brain	80.91 ± 2.72	90.82 ± 4.11	66.72 ± 5.71	0.7872 ± 0.0341
GM	**83.18 ± 2.08**	**91.92 ± 4.36**	**70.56 ± 5.04**	**0.8124 ± 0.0248**

**Table 15 brainsci-10-00181-t015:** Classification performance of different methods for T1 task.

Model	*ACC* (%)	*SEN* (%)	*SPE* (%)	*AUC*	*P*
Islam et al.	80.21 ± 3.26	88.85 ± 3.06	70.00 ± 4.57	0.7943 ± 0.0354	<0.001
H. Wang et al.	83.56 ± 3.38	81.92 ± 6.87	84.13 ± 4.73	0.8304 ± 0.0424	=0.008
Yuan et al.	80.21 ± 2.83	82.31 ± 4.61	77.73 ± 4.49	0.8002 ± 0.0314	<0.001
proposed	**87.50 ± 2.08**	**89.62 ± 3.86**	**84.99 ± 3.39**	**0.8731 ± 0.0219**	

**Table 16 brainsci-10-00181-t016:** Classification performance of different methods for T2 task.

Model	*ACC* (%)	*SEN* (%)	*SPE* (%)	*AUC*	*P*
Islam et al.	77.73 ± 3.34	91.16 ± 3.21	58.33 ± 6.48	0.7478 ± 0.0445	<0.001
H. Wang et al.	80.68 ± 1.83	**91.92 ± 4.69**	64.45 ± 7.53	0.7818 ± 0.0232	=0.014
Yuan et al.	79.32 ± 2.14	89.62 ± 6.89	64.44 ± 6.19	0.7703 ± 0.0358	<0.001
proposed	**83.18 ± 2.08**	**91.92 ± 4.36**	**70.56 ± 5.04**	**0.8124 ± 0.0248**

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
