# Peer review of "Multi-View Based Multi-Model Learning for MCI Diagnosis"

_brainsci, 2020, doi:10.3390/brainsci10030181_

Round 1

Reviewer 1 Report

The authors have substantially addressed all my concerns. The clarity of the manuscript is improved, even if minor editing is yet recommended. However, the manuscript can be accepted for publication.

Reviewer 2 Report

recommend accepting with minor grammatical revisions

This manuscript is a resubmission of an earlier submission. The following is a list of the peer review reports and author responses from that submission.

Round 1

Reviewer 1 Report

This paper presents an interesting methodology to detect Alzheimer's disease onset within a deep learning framework. The novelty of the proposed approach relies on the use of a combined analysis of 2D and 3D brain MRI scans. Despite this being the novelty of this work, I am worried its fundamental basis could be questionable, as I am going to discuss in detail in the following. Besides, the statistical methodology should be clarified. There are some minor issues about the writing quality. I think this work could be of potetntial interest for the journal's readership and I would recommend its publication provided that a major revision is suggested.

Paper foundations. I think the underying hypothesis of this work are questionable, especially looking at the "Introduction" seciton. It seems to me that the authors failed to outline some important aspects of Decision Support Systems designed for Alzheimer's disease and recent works were missed. In particular, it is not clear why a combination of 2D and 3D images should be considered. I don't agree with the idea that "3D images contain a lot of redundant information" in order to justify the use of 2D images, which are used WITH 3D images, thus not removing this (questionable) redundancy. Moreover, if a general framework should be invoked about AD classification I'd rather mention voxel-based [1,2], ROI [3] and hybrid approaches [4,5]. I think these recent works should deserve a mention. However, even accepting this hypothesis (3D redundancy), I think the author should confirm it by measuring what happens, in terms of accuracy, when using only 3D, only 2D and then their combination. Finally, I am really concerned by the statistical robustness of the proposed analysis. There are two major points I would recommend to clarify or improve in order to improve the overall quality of this study. On one hand, I cannot understand how many cross-validation rounds were performed, however a single ten-fold cross-validation is not enough. Especially because in this way it is not posssible to measure the accuracy uncertainty and no decisive conclusion can be drawn about the performance. On the other hand, I think that whatever being the decision support system implemented, the clinical interpretation of the results should be kept as an indirect soource of validation. In this sense I would like to know if the proposed methodolgy can be used to rank the different brain regions in relation with the diagnosis (a sort of feature importance measure) and what are the brain regions detected by their model.

[1] Zhang, Feng, et al. "Voxel-based morphometry: improving the diagnosis of Alzheimer’s disease based on an extreme learning machine method from the ADNI cohort." Neuroscience 414 (2019): 273-279.

[2] Nanni, Loris, et al. "Texture descriptors and voxels for the early diagnosis of Alzheimer’s disease." Artificial intelligence in medicine 97 (2019): 19-26.   [3] Wang, Yun, et al. "Diagnosis and prognosis of Alzheimer's disease using brain morphometry and white matter connectomes." NeuroImage: Clinical 23 (2019): 101859.   [4] Amoroso, Nicola, et al. "Multiplex networks for early diagnosis of Alzheimer's disease." Frontiers in aging neuroscience 10 (2018): 365.

[5] Gupta, Yubraj, et al. "Early diagnosis of Alzheimer’s disease using combined features from voxel-based morphometry and cortical, subcortical, and hippocampus regions of MRI T1 brain images." PloS one 14.10 (2019): e0222446.

Reviewer 2 Report

The authors present a deep-learning framework architecture mixing 2D and 3D information for AD / MCI / NC classification using ADNI data. Models are compared and tested in a 10 fold cross-validation setting with an ablation study.  

While the proposed architecture is interesting, different aspects of the paper could be improved.

Authors are kindly requested to check the manuscript thoroughly for style and grammar mistakes and should avoid as much as possible the use of colloquial sentences (no sentence beginning by And, no sentence containing only a subordinate...). These imperfections make the manuscript more difficult to follow and down weight the interest of the paper.

Introduction

Regarding the content, authors should be aware that there is currently no drug for the disease. Therefore the sentence: "

If MCI is detected and intervened in time, it can delay the progression of Alzheimer’s disease.

"

should be removed or specific references should be added. The MCI group may be extremely heterogeneous in terms of position along the disease axis. Instead of a classical AD vs MCI dichotomy, research is now more focused on the distinction between stable and converting MCI problem with a much stronger clinical implication.

Recent works on such classification methods using deep learning have been published and should probably be mentioned in the introduction (Basaia et al 2019 NeuroImage Clinical) with an accuracy of 85%

Methods:

The methods rely heavily on preprocessing techniques such as registration and segmentation. While ADNI is a very biased cohort in the sense that very limited vascular injury is present in the included subjects, in practice, WM contains a lot of additional information. Thus the sentence saying that WM has no information should be down-weighted. Furthermore, in the presence of WM lesions, GM segmentation are known to be impacted (Levy-Cooperman Stroke 2008). How would it be taken into account? Authors should discuss that issue and the question of generalisability to other cohorts than ADNI.

Indication of the registration tool use for resampling to standard space should be included.

Regarding the choice of 2D slices, there may be a strong bias in the selection as different slices may be selected across diagnostic groups. To be consistent, the same slices in the same order should be selected across subjects. Have the authors investigated any pre-existing bias due to the 2D slice selection?

The authors should specify what was optimised in the training: was it a multiclass cross-entropy, multiple binary CE?

The authors mention the class imbalance - Multiple works have been proposed to account for it - Could they add this small adjustment if this was indeed the cause of the decreased performance.

Could the authors specify the nature of the cross-validation- Was it stratified to ensure similar proportions across disease groups?

Results and experiments

The authors are advised to make adjustments to the table captions shown in the results. They should be more detailed to be able to understand the table if taken alone.

Since a cross-validation setting was adopted,  results should be presented as mean (sd) - Appropriate statistical tests should be performed to compare the methods

An interesting introspection of the result would be to check the association between cognitive score and level of misclassification to check if it reflects the continuity of the pathology - For instance one may expect that the NC misclassified as MCI have lower MMSE.

The authors state that it is essential to extract features (as grey matter maps) before feeding them to the network. It would be important to check that results are much deteriorated when feeding the raw images (registered to template space). It would save a lot of processing time.